# The effective family size of immigrant founders predicts their long-term demographic outcome: From Québec settlers to their 20th-century descendants

Damian Labuda[1,2]*, Tommy Harding[1,3], Emmanuel Milot[3], Hélène Vézina[4]

1 Centre de Recherche, CHU Sainte-Justine, Université de Montréal, Montreal, Québec, Canada,
2 Département de Pédiatrie, Université de Montréal, Montreal, Québec, Canada, 3 Département de chimie, biochimie et physique, Université du Québec à Trois-Rivières, Trois-Rivières, Québec, Canada, 4 Projet BALSAC, Université du Québec à Chicoutimi, Chicoutimi, Québec, Canada

* damian.labuda@umontreal.ca

**Data Availability Statement:** The genealogical data used in this study are deposited in the Dataverse: https://doi.org/10.5683/SP3/BKP6TW. They are available upon request from Hélène Vézina

## Abstract

Population history reconstruction, using extant genetic diversity data, routinely relies on simple demographic models to project the past through ascending genealogical-tree branches. Because genealogy and genetics are intimately related, we traced descending genealogies of the Québec founders to pursue their fate and to assess their contribution to the present-day population. Focusing on the female and male founder lines, we observed important sex-biased immigration in the early colony years and documented a remarkable impact of these early immigrants on the genetic make-up of 20th-century Québec. We estimated the immigrants' survival ratio as a proportion of lineages found in the 1931–60 Québec to their number introduced within the immigration period. We assessed the effective family size, EFS, of all immigrant parents and their Québec-born descendants. The survival ratio of the earliest immigrants was the highest and declined over centuries in association with the immigrants' EFS. Parents with high EFS left plentiful married descendants, putting EFS as the most important variable determining the parental demographic success throughout time for generations ahead. EFS of immigrant founders appears to predict their long-term demographic and, consequently, their genetic outcome. Genealogically inferred immigrants' "autosomal" genetic contribution to 1931–60 Québec from consecutive immigration periods follow the same yearly pattern as the corresponding maternal and paternal lines. Québec genealogical data offer much broader information on the ancestral diversity distribution than genetic scrutiny of a limited population sample. Genealogically inferred population history could assist studies of evolutionary factors shaping population structure and provide tools to target specific health interventions.

(Helene_Vezina@uqac.ca) or the BALSAC team (balsac@uqac.ca). The access request is to be made by completing the form on the BALSAC website (https://balsac.uqac.ca/acces-donnees/). This step is required due to ethical restrictions on the publication of genealogical information dating back less than 100 years.

**Funding:** Computations were made on the Cedar and Mammouth supercomputers at Simon Fraser University and the Université de Sherbrooke, respectively, managed by Compute Canada and Calcul Québec, funded by the Canada Foundation for Innovation, the Ministère de l'Économie et de l'Innovation du Québec and the Fonds de recherche du Québec – Nature et technologies. This research was supported by the Fonds de recherche du Québec – Santé, via le Réseau de médecine génétique appliquée (DL, HV), the National Sciences and Engineering Research Council of Canada - 06317 (DL), and by DL donations through the Fondation de l'Hôpital Sainte-Justine.

**Competing interests:** The authors have declared that no competing interests exist

## Introduction

Population genetics is about population structure, genetic history, and phylogenetic connections. Human evolution involves migrations, population splits, size fluctuations, founder effects, and admixture. To reveal populations' history, we collect actual genetic diversity data and model populations' ascending genealogical trees [1]. These, however, referred to as *gene trees*, habitually represent phylogenetic reconstructions back to prehistorical times. Most of these reconstructions rely on likelihoods using demographic models that assume long-term equilibria. However, as Alan Fix [2], paraphrasing James Neel [3], stated: "the history of the human species and particular populations suggests that long-term equilibria, an assumption of many genetic models, is only a convenient mathematical fiction." (see also [4]). In turn, genealogies represent an objective data source to learn about populations' demographic past and their ensuing genetic structure [5–11]. Therefore, studies of populations with an extensive genealogical record may fill the gap between historical reality and computer modeling [11]. However, many genealgies, like in the coalescent represent only ascending lines from sampled individuals back in time to the founders or most recent common ancestors. The genealogical coverage exists for a handful of human groups, such as small isolates, local scale data on religious groups [12–15] or insular populations [8,16].

The Québec population is presumably the largest one in this category. Its population started with the 17<sup>th</sup> century European settlements in the St-Lawrence Valley, primarily by immigrants from France [17]. BALSAC [18,19] is a genealogical database, which includes records on married individuals from all documented population founders down the generations, including lost and surviving lines. It is unique in its quality and breadth, with a quasi-complete reconstruction of the descending genealogies of Québec consisting of millions of individuals. These data allow studies on evolutionary and selection events [6,20,21], geographical patterns of disease-causing mutations [22,23] and many more. However, except for particular applications, BALSAC was never explored in its entirety to study the history of the population it represents. Maternal and paternal lines taken together link all generations of parents and children. They are easy to retrace, providing a simple picture of kin relations within this population [24]. When used to study autosomal or X-chromosome transmissions, the same genealogical network leads to innumerable combinations of uniparental genealogical pathways and is better suited to particular applications [21–23].

Genealogical lines relating mother-and-daughter or father-and-son mirror the transmission of the uniparentally inherited mitochondrial DNA (mtDNA) and the non-recombining portion of the Y-chromosome, respectively. They are particularly relevant to studying different aspects of populations' history. They reveal male and female contributions to the genetic diversity of extant populations due to sex bias in reproductive variance, sex-biased migration, matrilocality *vs.* patrilocality [25,26]. However, their joint analyses relying on their genetic diversities are complicated [26] because the mutation rates of mtDNA and the Y-chromosome are very different [27]. Such studies are further complicated by differences in the effective population size of males and females. Yet, these problems disappear once we follow maternal and paternal genealogical lines and their distribution among generations down the road [8]. We may concurrently analyze maternal and paternal lines using genealogical data, disregarding any mutational bias.

In this study, we use the descending genealogy of married Québec individuals to follow the fate of maternal and paternal lineages over a period covering more than three hundred years. Our goal is to assess how historical immigration waves shaped the contemporary Québec population and understand the demographic/genetic legacy of Quebec lineage founders as a function of their immigration time and the reproductive success of their descendants. Towards this

goal, we introduced novel statistics, only applicable to the descending genealogical data, to follow, with time passing, the proportion of lost and surviving lines. We show that EFS predicts the survival of the descendant lines over generations. This observation can be extended to the genetic contribution of the autosomes. Parents with high EFS left plentiful married descendants, putting EFS as the most important variable determining the parental demographic success throughout time for generations ahead. EFS of immigrant founders appears to predict their long-term demographic and, consequently, their genetic outcome. Québec genealogical data offer much broader information on the ancestral diversity distribution in the current population than genetic scrutiny of a limited population sample. Such data on genealogically inferred genetic population structure could become crucial in targeting specific health interventions [22,23,25]. Finally, our analysis of the entire bank of Québec descending genealogies should encourage further studies using BALSAC remarkable resources.

## Materials and methods

### BALSAC data

Genealogical information on the historical Québec population is digitalized in the BALSAC population database (http://balsac.uqac.ca) with a remarkably high degree of completeness [19]. BALSAC includes data from the 17th and 18th centuries in Québec that were initially compiled within the Early Québec Population Register of the Programme de recherche en démographie historique (PRDH) of the Université de Montréal [28] (https://www.prdh-igd.com/en/home). The genealogies were primarily reconstructed from Catholic church records about life-history events such as marriages, births, or deaths. Moreover, BALSAC and PRDH collected valuable information on genealogical connections beyond Québec, such as data on European and Acadian ancestors or emigrant marriages outside Québec (e.g., [7,29]). We define immigrants as the first generation of settlers to marry in Quebec. Most immigrants introduced new maternal and paternal lines, representing new mitochondrial DNA and Y-chromosome lineages, respectively, and are considered lineage founders. However, some immigrants were born in Quebec to parents who married before coming to Québec. Also, some were related to each other upon their arrival in Québec (e.g., siblings, cousins). In those cases, using BALSAC data from outside of Québec, the most recent common ancestor (maternal or paternal) of these related immigrants were identified as the lineage founder to avoid counting a lineage more than once (S1 Fig). The population of married couples between 1931 and 1960 was set as the reference for the 20th-century population of Québec. The 30-year period approximates the time of one generation [30].

### Demographic parameters

From BALSAC, we extracted all married individuals and categorized them either as immigrants or as Québec-born (QB) descendants of the immigrants (c.f. S1 Fig). Immigrant males and females were counted independently to separate maternal and paternal lineages and pooled into waves $w_i$ corresponding to their marriage years $t_i$, starting in 1621 and up to 1930. According to their marriage year $t_i$, QB were grouped into corresponding strata $s_i$. $N_i$ represents the count of immigrants from wave $w_i$ in the year $t_i$ (both sexes, or only either males or females, according to the context). $Nw_i$ denotes the number of their male or female descendants married between 1931 and 1960 (here, only maternal lineages in married women are counted: male mtDNA carriers are excluded because they do not transmit mtDNA further). The ancestors of $Nw_i$ are referred to as *contributing* individuals: *contributing* immigrants $Nc_i$ and *contributing* Québec-born $Nc_{QBi}$. Consequently, $N_{QBi}$ represents the count of $s_i$ QB married in year $t_i$ and $Ns_{QBi}$ that of their descendants married in Québec between 1931–1960.

To assess the overall rate of the demographic growth $k$ in Québec, we used historical estimates of the Québec population size from Larin [31], census reports [32], and BALSAC data (only considering married individuals). In a growing population, $N_t = N_{t-x}(1+k)^x$, where $N_t$ is the population size at the year $t$, $N_{t-x}$, its size at $x$ years earlier, and $k$ is the population growth rate per year. Because $k$ is small, $(1+k)^x \sim e^{kx}$, it follows that $\ln(N_t)-\ln(N_{t-x}) = k \cdot x$, such that the population growth rate $k$ can be readily estimated from the slope of $\ln N_t$ versus time $t$ (S2 Fig).

## New descriptors of descending genealogies

Remarkably, descending genealogies come with additional information not captured in the ascending genealogies. It includes the number of initial founders and their descendants; the surviving (and lost lineages). We introduce three measures to describe this information on population demographic history, namely: (i) the survival ratio ($SR_i$), (ii) the net growth ($dc_i$), and (iii) net extinction ($a_i$) rates. $SR_i = Nc_i/N_i$ for immigrants and $Nc_{QBi}/N_{QBi}$ for Québec-born individuals, thereby describing the proportion of contributing individuals within $w_i$ or $s_i$. The net growth rate $dc_i$ and extinction rate $a_i$, between historical $t_i$, and the descendant population of the 1931–60 Québec is estimated using $t = 1947.5$ ($t_{1947.5}$) as an average target year of marriages between 1931-60. Both $dc_i$ and $a_i$ are evaluated using the same rate equation we used to estimate $k$, except that they only describe a net growth or extinction of the population fraction from time $t_i$ down to the average $t = 1947.5$. Thus, we write $Nw_i = Nc_i(1+dc_i)^{(t_{1947.5}-ti)}$, whereby $dc_i = \ln(Nw_i/Nc_i)/(t_{1947.5}-t_i)$ and, likewise, $dc_{QBi} = \ln(Ns_{QBi}/Nc_{QBi})/(t_{1947.5}-t_i)$. As for the net extinction rate for immigrants: $Nc_i = N_i(1-a_i)^{(t_{1947.5}-ti)}$ and $a_i = \ln(N_i/Nc_i)/(t_{1947.5}-t_i)$ and likewise for the Québec-born $a_{QBi} = \ln(N_{QBi}/Nc_{QBi})/(t_{1947.5}-t_i)$. Note that $dc_i$ and $a_i$ complement $SR_i$ and $1-SR_i$, respectively, and consider the time factor ($t-t_i$). We pooled data in bins of five or ten years to plot the data to reduce the variance, yet the values shown are averages per year.

## Progeniture

We refer to the number of married children of a couple as their effective family size (EFS) [33] or the number of potentially fertile children (i.e., grand-child bearing children) [6,8,33]. While BALSAC data does not allow to calculate Darwinian fitness of individuals directly, fortunately, the EFS is strongly correlated to fitness in the French-Canadian population and was used here as a proxy [6,33]. However [34], see the danger of conflating selection and inheritance with such a fitness proxy, although the strong correlation limits this problem. We measured EFS sex-wise by counting the number of married daughters (EFS-daughters) or sons (EFS-sons) for mothers and fathers, respectively, or pooling both sexes' maternal and paternal children. We separately examined the EFS of contributing and non-contributing parents from immigrants or their Québec-born descendants.

## Genetic contribution

In genealogical studies, genetic contribution (GC) describes the expected number of copies of a neutral allele contributed by a population founder to their descendants [11,35,36]. By default, it relates to the autosomal inheritance, such that in a complex genealogy, a founder allele can be transmitted through different paths, all of which have to be taken into account when computing GC [16]. Estimation of the contribution of uniparentally inherited maternal or paternal lineages is straightforward. We count the number of lines attributed to every contributing founder ($Nw_i$ above). Maternal lines follow the mother-to-child path, propagated further through her daughters and ending at each son, and paternal lines follow father-to-son transmission. Conversion of counts to frequencies gives relative contribution of uniparental lines founders, tantamount to frequencies of the corresponding mtDNA and Y-chromosomal

founder lineages. Estimating "autosomal" GC explores the same genealogical paths, exploring all their combinations because they are crossed down, with a probability of transmission of 0.5 from a parent to a child, disregarding their sex. In other words, in contrast to maternal and paternal lines only, an autosomal founder allele can descend to a target population through all possible maternal and paternal pathways linking the founder to its descendants. Autosomal GC was estimated using the *gen.gc* function of the R package GENLIB [37]. Genealogical error rate, resulting from false paternity or maternity (e.g., due to undeclared adoption) and errors in parish records and their transcription is estimated to be below 1% [24]. Given the amount of genealogical data we processed, and the nature of our findings, errors at such a level can be neglected, as they are not expected to affect our results and conclusions.

# Results

## Population founders and their descendants in 1931–60 Québec

The European founding of the Québec population started with the establishment of Québec City in 1608, followed by urban centers at Trois-Rivières (1634) and Montreal (1642), upstream of the Saint Lawrence River. The settlements in Beaupré (1636) and Baie-Saint-Paul (1673), north-east of Québec City, largely contributed to the peopling further downstream. In 1763, the Treaty of Paris sealed the British conquest of Québec and Canada. Around 1800, the Québec population had grown to about 220,000 people [31], eventually reaching 2,874,662 in 1931 [32]. In BALSAC (version as of March 2015), we find genealogies of 3,340,072 individuals married in Québec before 1961 (2,018,038 before 1931). This includes 455,687 immigrants, of whom 276,946 settled before 1931. Of the latter, 29,668 immigrant women and 32,608 immigrant men contributed to the pool of uniparentally-inherited lines of 1931–60 Québec (Table 1). In its early years, the Québec population primarily grew due to the inflow of immigrant settlers (Fig 1A and Table 1 and S2 Fig). The first marriage documented in Québec Catholic registers took place in 1621.

## Historical periods and population expansion

Based on historical characteristics, we divided the European settling of the Quebec territory into five immigration periods. From 1621 to 1680, the first period was marked by the arrival of

**Table 1. Numbers of immigrants and Québec-born individuals who married in Quebec in five historical periods.** Numbers of distinct and novel (newly introduced, i.e., in addition to those already present and introduced earlier), uniparentally inherited lineages (separately maternal and paternal) are reported as well. Contributing immigrants are those who transmitted their descendant lineages in the 1931–60 population.

| Periods | 1620–1680 | | 1681–1750 | | 1751–1800 | | 1801–1850 | | 1851–1930 | | Sum down to 1930 | | 1931–60 | |
|---|---|---|---|---|---|---|---|---|---|---|---|---|---|---|
| | Women | Men | Women | Men | Women | Men | Women | Men | Women | Men | Women | Men | Women | Men |
| **Immigrants** | 1241 | 1631 | 1580 | 4827 | 3981 | 6766 | 26749 | 27515 | 99518 | 103138 | 133069 | 143877 | 81565 | 97176 |
| Distinct lineages | 1081 | 1520 | 988 | 4101 | 2567 | 5345 | 22781 | 24154 | 75553 | 86899 | 102094 | 121221 | 61524 | 79543 |
| Novel lineages | 1081 | 1520 | 930 | 4052 | 2543 | 5279 | 22696 | 24030 | 74844 | 86340 | 102094 | 121221 | 57961 | 76357 |
| **Contributing immigrants** | 706 | 827 | 470 | 1753 | 922 | 1816 | 2117 | 2344 | 25453 | 25868 | 29668 | 32608 | 81565 | 97176 |
| Distinct lineages | 616 | 793 | 238 | 1440 | 331 | 1281 | 1914 | 2106 | 20923 | 22777 | 23873 | 28233 | 61524 | 79543 |
| Novel lineages | 616 | 793 | 206 | 1416 | 321 | 1255 | 1878 | 2046 | 20852 | 22723 | 23873 | 28233 | 57961 | 76357 |
| **Quebec-born** | 312 | 56 | 14726 | 11135 | 43526 | 37891 | 146715 | 134197 | 707013 | 645521 | 912292 | 828800 | 586049 | 557244 |
| **Contributing Quebec-born** | 286 | 49 | 13987 | 10272 | 42871 | 36197 | 144000 | 130520 | 689709 | 630727 | 890853 | 807765 | 580335 | 553516 |
| **Immigrants + Quebec-born** | 1553 | 1687 | 16306 | 15962 | 47507 | 44657 | 173464 | 161712 | 806531 | 748659 | 1045361 | 972677 | 667614 | 654420 |
| **Fraction of immigrants** | 0.80 | 0.97 | 0.10 | 0.30 | 0.08 | 0.15 | 0.15 | 0.17 | 0.12 | 0.14 | 0.13 | 0.15 | 0.12 | 0.15 |

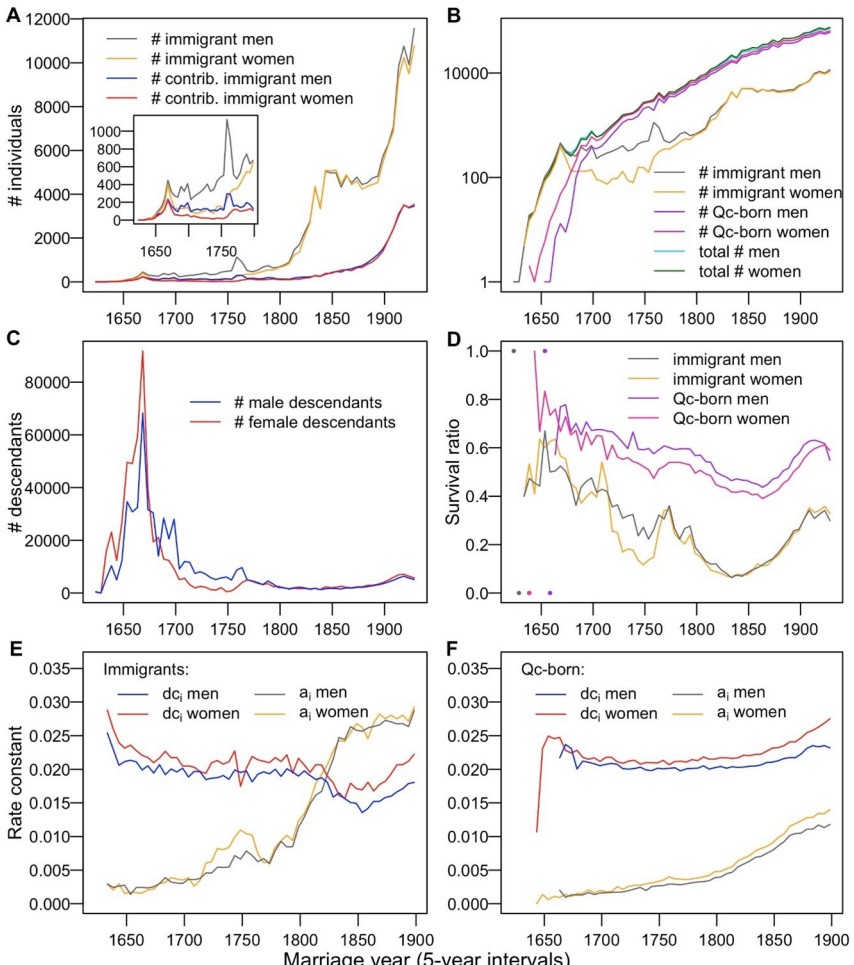

**Fig 1. Immigrants, their Quebec-born descendants and related statistics as a function of time.** The number of immigrants and their contributing portion at linear scale (inset shows early history at a magnified scale) (**A**). The number of all immigrants, Québec-born and their total in logarithmic scale (**B**). The number of immigrants' descendants, $Nw_i$, within the 1931–60 population as a function of immigrants' settlement time (**C**). Evolution of the survival ratio SR of immigrants and Quebec-born (**D**). Corresponding plots of the net growth ($dc_i$) and net extinction ($a_i$) rates for the immigrants (**E**) and the Quebec-born (**F**).

the Filles du Roy in the years 1663–73. They were French women whose immigration to New France was sponsored by King Louis XIV (thus "King's daughters"), designed to boost New France's population. They represent the first immigration peak in Fig 1A. About twelve hundred women and sixteen hundred men immigrants were registered at that period. Out of these, 708 women and 830 men left descendant lineages (Table 1) that account for 66.6% of maternal and 47.3% paternal lineages in the 1931–60 Québec population l (Table 2).

The second immigration cycle in Quebec, from 1681 to 1750, took place after the arrival of Filles du Roy and before the British conquest (1760). This period contributed 13.9% and 30.6% of maternal and paternal descendants to the 1931–60 population. The paternal contribution prevailed over the maternal lineages. Altogether, the uniparentally inherited lines introduced by the immigrants before 1751 are found in ~467,000 women and ~431,000 men married between 1931 and 1960 (80.5% and 77.9% of all married descendants from before 1931, respectively; Table 2). At the turn of the 18th century, QB individuals surpassed the immigrants in numbers and dominated the bulk of the population (Fig 1B).

**Table 2. Contribution of immigrants from different periods to the 1931–60 Québec, both in the number of their married descendants and in the number of distinct maternal and paternal lineages.**

| Contributing Periods | | Married | | Lineages | | 1931–60 descendants / lineage | |
|---|---|---|---|---|---|---|---|
| **1931–60** | | Women | Men | Maternal | Paternal | Maternal | Paternal |
| | Counts | 87 279 | 100 904 | 61 526 | 79 545 | 1.4 | 1.3 |
| | % 1931–60 | 13.1 | 15.4 | | | | |
| **All before 1931** | counts | 580 335 | 553 516 | 23 873 | 28 233 | 24.3 | 19.6 |
| | % 1931–60 | 86.9 | 84.6 | 28.7 | 26.7 | | |
| | % <1931 | 100.0 | 100.0 | 100.0 | 100.0 | | |
| **1851–1930** | counts | 63 623 | 54 917 | 20 923 | 22 777 | 3.0 | 2.4 |
| | % 1931–60 | 9.5 | 8.4 | 25.2 | 21.5 | | |
| | % <1931 | 11.0 | 9.9 | 87.6 | 80.7 | | |
| **1801–1850** | counts | 17 575 | 17 513 | 1 914 | 2 106 | 9.2 | 8.3 |
| | % 1931–60 | 2.6 | 2.7 | 2.3 | 2.0 | | |
| | % <1931 | 3.0 | 3.2 | 8.0 | 7.5 | | |
| **1751–1800** | counts | 31 675 | 5 064 | 331 | 1 281 | 95.7 | 39.1 |
| | % 1931–60 | 4.7 | 7.7 | 0.4 | 1.2 | | |
| | % <1931 | 5.5 | 9.0 | 1.4 | 4.5 | | |
| **1681–1750** | counts | 80 923 | 169 272 | 238 | 1 440 | 340.0 | 117.6 |
| | % 1931–60 | 12.1 | 25.9 | 0.3 | 1.4 | | |
| | % <1931 | 13.9 | 30.6 | 1.0 | 5.1 | | |
| **1621–1680** | counts | 386 539 | 261 750 | 616 | 793 | 627.5 | 330.1 |
| | % 1931–60 | 57.9 | 40.0 | 0.7 | 0.8 | | |
| | % <1931 | 66.6 | 47.3 | 2.6 | 2.8 | | |

From 1751 to 1800, the third colonization period was marked by a remarkable peak of male immigrants (Table 2 and Fig 1A) due to the demobilization of French soldiers [31], who settled in Québec after the British takeover in 1760. They were joined by descendants of French pioneers from Acadia (present-day Nova Scotia and New Brunswick), who were survivors of the British deportation campaign of 1755. Acadian settlers were followed by British Loyalists fleeing the United States after the US Declaration of Independence. Immigration dramatically increased after 1800 (Fig 1A and 1B). It slowed down in the middle of the 19th century and resumed at the turn of the 20th century. About 55,000 immigrants arrived in Quebec during the 1801–1850 period (4th period) and more than 200,000 from 1851 to 1930 (5th period).

Up to 1680, the Québec population growth mainly reflects an increasing number of new settlers. Estimates of $k$ for this period, based on Larin's data ($k = 0.084$) and BALSAC data ($k = 0.11$), concord very well with each other, considering the relatively small number of individuals involved (S2 Fig). During the 1681–1850 period, the population growth rate is estimated at 0.026 with both data sets (Larin and BALSAC) and $k \sim 0.015$ following 1850. The population growth after 1680 is primarily due to QB, while the contribution of the new immigrants dropped to about 15% (Fig 1B). In comparison, the average population growth rate in Europe between 1500 and 1900 was about 0.006 per year [38].

## Survival ratio, net growth, and extinction of lineages

While focusing on maternal and paternal lines, we counted how many lineages survived, how many were lost, how fast they grew, and what was their extinction rate; all this relative to the 1931–60 Québec population. The demographic outcome of an immigration wave $i$ among the 1931–60 descendants, $Nw_i = N_i[(1 - a_i)(1 + dc_i)]^{(t_{1947.5} - t_i)}$ is a function of the immigrants'

number $N_i$, their arrival time $t_i$, the extent of their growth ($dc_i$), and line loss ($a_i$). The maximum $Nw_i$ coincides with the immigration peak of the Filles du Roy (Fig 1C). Note that an excess of females $Nw_i$ preceded and was within this maximum. This was followed by much greater male than female $Nw_i$, reflecting the prevalence of male immigrants. In contrast to the 17th-18th century immigrant contributions (Tables 1 and 2), the $Nw_i$ of the 19th-century immigrants (Fig 1C) appears strikingly low for both sexes, despite an upsurge in immigrant numbers (Fig 1A and 1B) as judged by the relatively low level of descendants in 1931–60 Québec (S3 Fig).

Before 1681, the survival ratio of immigrants was around 0.6 (Fig 1D). Its subsequent decrease can be associated with a drop in net growth, $dc_i$, and an increase in net lineage extinction, $a_i$. In the first half of the 19th century, there was a sudden $dc_i$ drop and a sharp rise in the immigrant $a_i$. This combination explains the minimum immigrant survival ratio with a nadir of 0.1 at around 1835. Around 1825, $a_i$ became larger than $dc_i$, (i.e., $\ln(N_i/Nc_i) > \ln(Nw_i/Nc_i)$, such that $N_i > Nw_i$). It means that from that time on, the number of immigrants exceeded the number of their descendants among the 1931–60 married (compare data in Tables 1 and 2).

## Effective family size—determinants of the outcome

The demographic outcome of immigrants depends on their reproductive performance (reproductive value [11]) and that of their descendant generations. The QB survival ratio follows a similar trajectory but with higher survival ratios and ranges from about 0.8 to a minimum of only 0.4 in the middle of the 19th century (Fig 1D). The extinction rate $a_i$ of QB paralleled that of the immigrants, albeit at a lesser pace. QB appear more demographically stable but still succumb to the same external factors, decreasing the survival ratio and slowing their growth. QB did not go through the 19th-century $dc_i$ downfall that affected immigrants (Fig 1E and 1F). EFS values are the highest for married individuals in the 17h-century. They are followed by a minimum around 1750, a rebound at the turn of the 19th century to eventually decline. Contributing QB and immigrants show similar ups and downs in EFS (Fig 2). However, overall, QB have a greater EFS than the immigrants (Fig 2 –middle curves). Non-contributing individuals had a much lower EFS throughout the study period. Immigrants showed the lowest EFS with an average of ~0.5, i.e., one married child per couple, either a boy or a girl (Fig 2).

Notably, the survival ratio of immigrants follows their EFS (Fig 3A and 3B). The survival ratio of the maternal and paternal lineages of the immigrants also strongly correlates with their corresponding EFS ($R^2$ = 0.91 and 0.90, respectively; p << 0.001). In other words, the proportion of the contributing immigrants' lineages from any time $i$ mirrors their EFS. This correlation also stands when we consider 1-year rather than 5-year intervals, despite a larger variance (S4 Fig). An immigrant with an EFS = 1 (1 child, either daughter or son) predicts that its descending lineage (maternal or paternal) has a ~0.3 probability of surviving up to ≥1931 (insets in Fig 3A and 3B). The correlation between the survival ratio and EFS is perpetuated over QB generations descending from these immigrants (Figs 3C and 3D and S4), although the correlation is less straightforward ($R^2$ = 0.65 and 0.69). And indeed, by the end of the 18th century, QB EFS and SR plots start to diverge and then cross by the end of the 19th century, thereby inverting the relation (see Discussion).

## Sex-asymmetry among immigrants

During the French Regime, there were fewer immigrant women than immigrant men. Before 1681, the immigrants' sex ratio (men to women) was 1.3 (Table 1). It went up to higher values in the middle of the 18th-century (S5 Fig) with an average of 2.1 between 1681 and 1800. At the colony's beginning, women married much younger than men and remarried more often

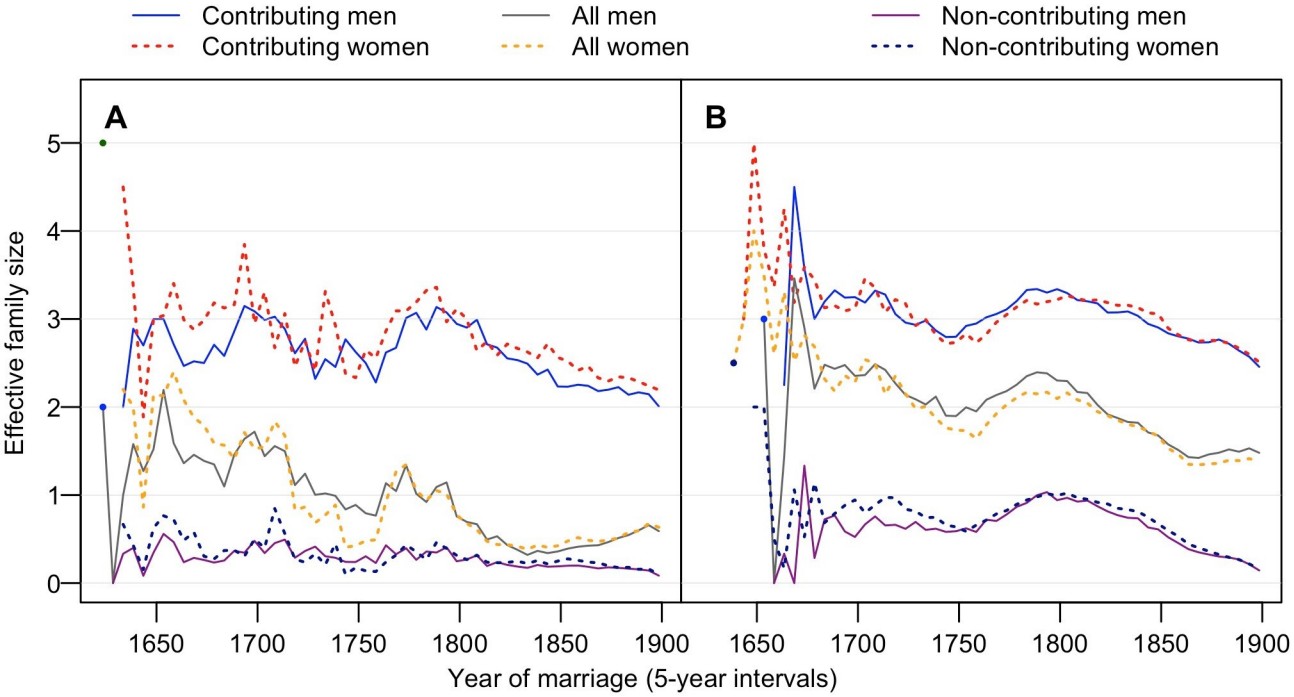

**Fig 2. Effective family size of maternal and paternal lineages at different time points.** Families of immigrants (**A**), and Quebec-born families (**B**), showing EFS of contributing, non-contributing descendants and their total (All). Maternal EFS-daughters are shown in dotted lines and paternal EFS-sons in solid lines.

[17,39]. In Fig 4, we show that before 1681 remarriages of women prevailed and that this tendency reverted after 1681. During the following years, given the shortage of immigrant women, immigrant men married Québec-born daughters of previous immigrant couples. This created a disequilibrium between EFS-daughters and EFS-sons of QB parents. According to BALSAC with the PRDH data included (see M&M), during the early time of Quebec (parental marriages before 1700), Québec-born males were less nuptially successful than their sisters (Table 1), as shown by the parents' EFSs that was lower than EFSd (Figs 2 and S6). It is plausible that many QB males were lost from the "radar" of Québec church records; as explorers of new territories themselves, they became emigrants.

Consequently, we observe the preferential impact of the matrilineal lineages from before 1681 (Fig 1C) because successive immigrant males marrying QB daughters promoted the expansion of maternal lines introduced earlier. (Note that years on the *x*-axis identify parental marriages, yet the statistics of the reporting children marriage, such as EFS, are taking place one generation later). At the same time, these male immigrants enriched the paternal lineages' genetic pool relative to that of the ill-represented immigrant maternal lines (Figs 1C and S7). In the 1931–60 population, we find 630 individuals per single maternal lineage introduced before 1681, compared to only 330 per single patrilineal line (Table 2). Considering all descendants from before 1801, we find three times more paternal than maternal lineages in the 1931–60 population, 3514 and 1185, respectively.

In contrast, over the 19th century, down to 1930, there were 24 883 paternal and 22 837 maternal contributing immigrants (Table 2). The sex ratio of 1.09 concords with an excess of male remarriages compared to the period before 1681 when female remarriages prevailed (ratio of 0.92). Male immigration bias profoundly affected the frequency spectra of male and female lineages introduced before 1801 (S7 Fig). Lower female lines diversity was observed at

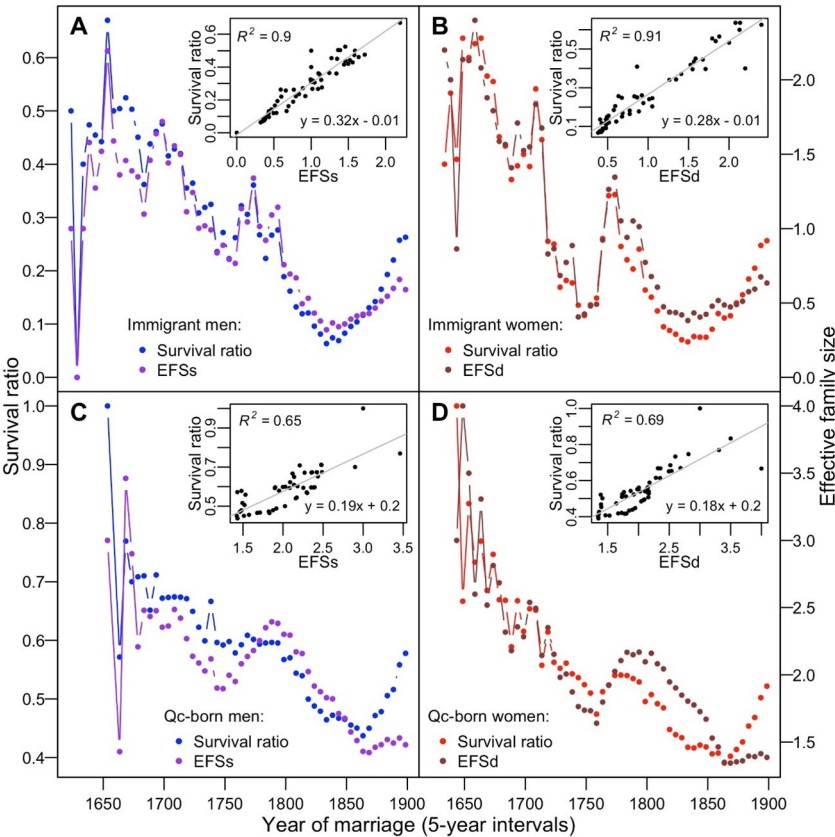

**Fig 3. Correlation between effective family size and survival ratio.** Overlapping plots of total EFSs and EFSd and their survival ratios for the immigrants (**A** and **B**) and the Quebec-born lineages (**C** and **D**). Their correlation plots are shown in the insets.

the molecular level by contrasting the Y chromosome versus mtDNA diversities in Gaspésie [40]. That is also known from the demographic comparisons of the frequency of patronyms and matronyms [41].

Differences in the contribution of the immigrant women and men as a function of settlement time and recorded through uniparentally inherited lines also apply to their respective "autosomal" GC's. We find that the contribution of women and men immigrants' maternal and paternal lines, respectively (Fig 1C), reflects that of their "autosomal" GC (Fig 5). Slight differences between uniparental and "autosomal" GC plots can be ascribed to all immigrants being considered autosomal founders. In contrast, some maternal or paternal founders were shared among related immigrants (see M&M). Computing "autosomal" GC assumes an equal probability of transmission of one of the two parental alleles, disregarding the variance due to linkage within the inherited chromosomal segments [10,36]. It is thus not unexpected that "autosomal" GC ideally correlates with the uniparental contributions (Fig 5). If genuine autosomal genetic diversity data were used, we would observe a significant variance [10] in the autosomal plots due to linkage, selection or admixture.

## Discussion

Turning to the coalescent [42], we can use the existing genetic variation to reconstruct ascending genealogical and phylogenetic trees modeling the past based on today's outcome. Such

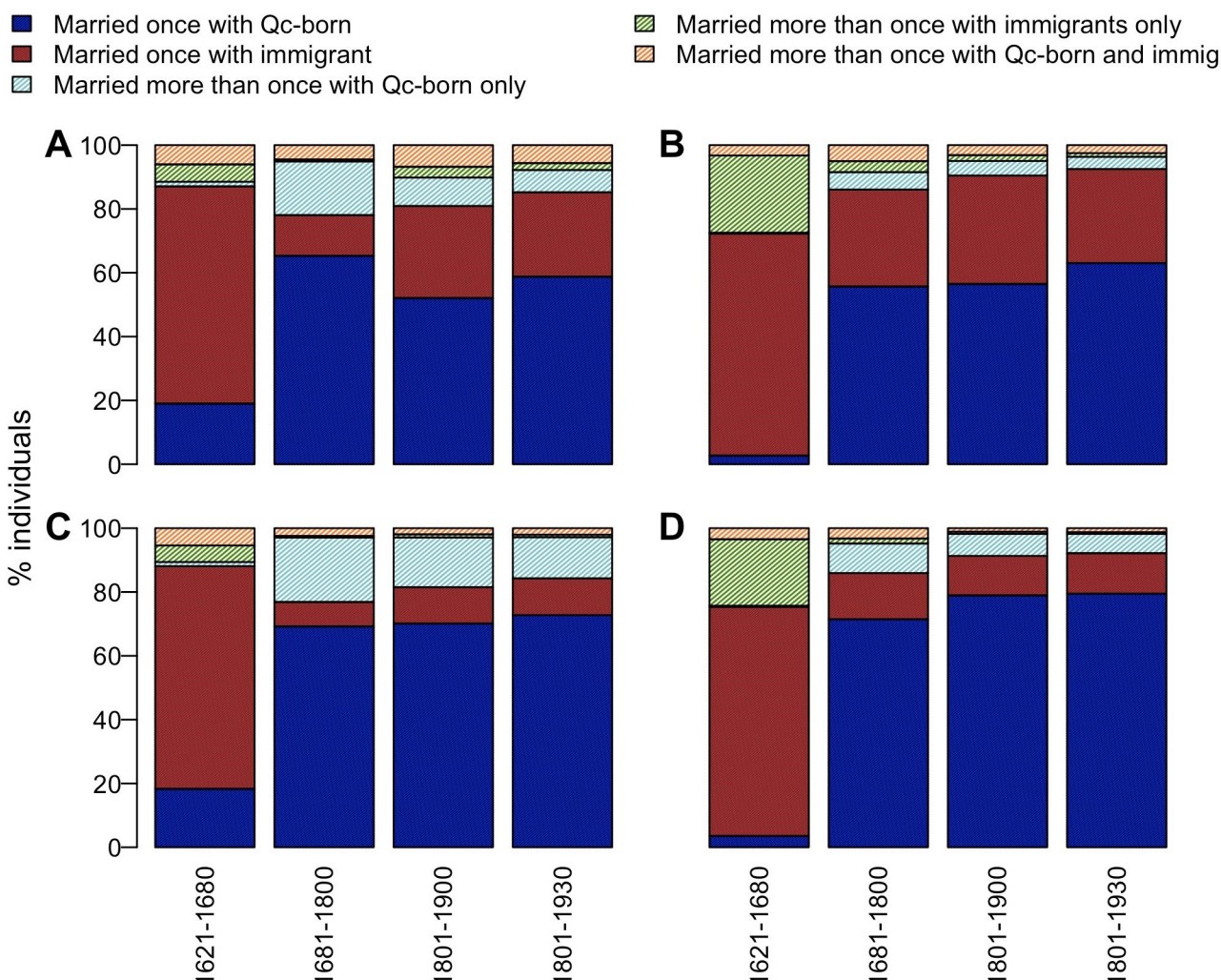

**Fig 4. Distribution of marriages and remarriages among marrieds at four time periods.** Different categories of marrieds are specified in the above histograms by their corresponding color codes. The contributing male and female immigrants are in **A** and **B**, respectively, whereas all (immigrants and Quebec-born) married men (**C**) and women (**D**).

historical projections ignore lines not represented within the genetically scrutinized sample because these lines already died out or simply escaped sampling. However, having access to population-wide descending genealogies, we do not have to depend on demographic models to infer history from the present-day diversity [1]. The BALSAC database contains digitized vital records from Quebec for over three centuries, providing a quasi-complete reconstruction of the descending genealogies of Québec's historical population. Moreover, genealogical errors, such as recording oversights, false paternity or maternity, are rare in Québec [19,24]. BALSAC information enables us to investigate the demographic past of the Québec population to understand the present better. Our study analyzes Quebec's uniparentally inherited maternal and paternal lines that parallel mtDNA and Y-chromosome inheritance much more than earlier studies [7,20,24]. We followed their fate from the origins of the Quebec colony (1608) down to the years 1931–60. As a result, we provided a broader perspective on the creation of the Québec population, extending the analysis of its immigration history well after the British Conquest in 1760 and by investigating factors and mechanisms prompting its demographic

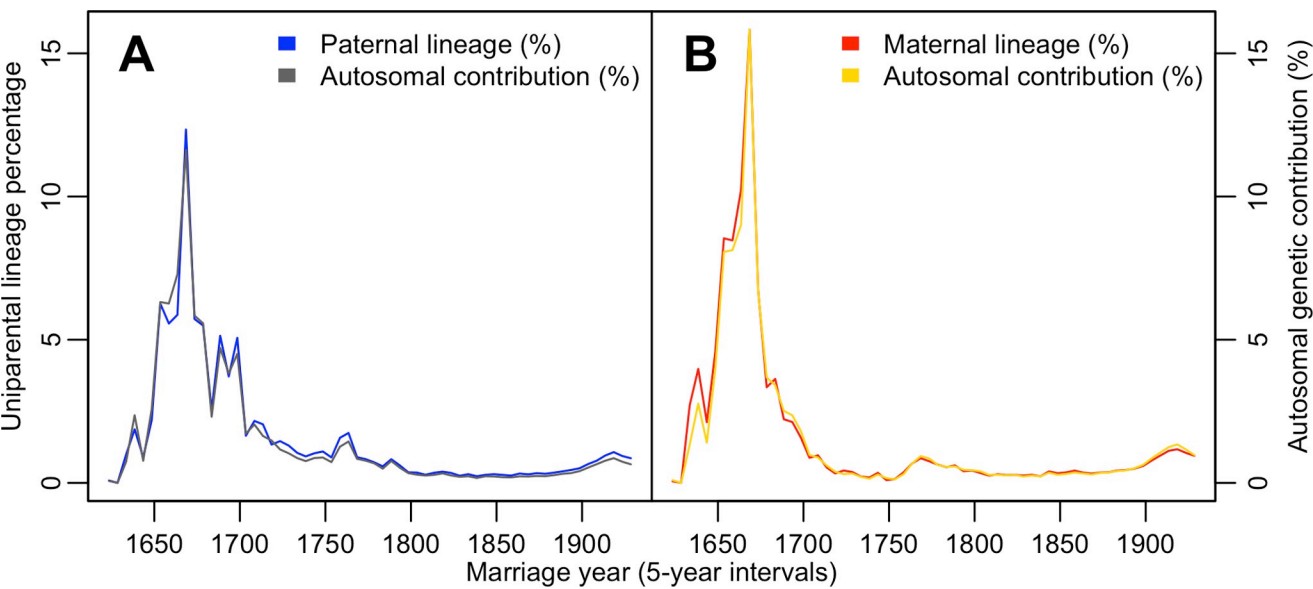

**Fig 5. Comparison of "autosomal" GC of immigrant women and men with GC of their maternal and paternal lines ($Nw_i$) within the 1931–60 population.** The fraction of the immigrant contribution ($y$-axis) is expressed per year averaged over five years ($x$-axis).

and geographical expansion. Our study differs from previous analyses using ascending genealogies of existing populations [7,35] or from the studies focused on descending data of a single or selected lineage [6,20,43]. It also differs from studies analyzing data sets from a limited number of parishes [13,44], religion groups [45] or geographic isolates [16,46] and an earlier study of uniparentally inherited lines in the Icelandic population [8].

We evaluated the yearly number of immigrants $N_i$ and their contributing part $Nc_i$, who left $Nw_i$ carriers of their uniparental lines among 1931–60 married descendants. We introduced new descriptors based on the knowledge of the descendant lines. First is the survival ratio, $SR_i = Nc_i/N_i$, which compares the success of consecutive waves of immigration in leaving descendants (through maternal and paternal lines) among the 1931–60 married individuals (Fig 1D). Do not confound $SR_i$ with the wave $i$ numerical outcome $Nw_i$, i.e., the number of descendant copies among 1931–60 marrieds (Fig 1C). We also followed the EFS along all maternal and paternal lines. To tell apart the role of fertility, represented by EFS, from extinction (due to drift or emigration), we introduced additional descriptive measures, the net growth $dc_i$ and the net extinction rate, $a_i$. They estimate net growth or net extinction of distinct lines, respectively, between a time point $i$ in the past and an extant population, such as that of the 1931–60 Québec (Fig 1E and 1F).

We observed that during most of the history of Québec, the $SR_i$ of the immigrants and QB individuals was declining (Fig 1D). $SR_i$ represents the proportion of the contributing lineages. At first glance, while reasoning in terms of a simple demographic model of a population of a constant size, we found this time dependence of $SR_i$ striking. Under genetic drift and in a growing population, $SR_i$ is expected to increase with time. Genetic drift [47] is more potent when a population is small [48] and is thus likely to reduce the number of lineages more effectively at the start of a colony. In general, genetic drift gets weaker in a growing population [49,50], implying that the proportion of surviving lineages introduced closer to the present should be greater than that of the earlier lineages. However, we observed the opposite, although perhaps not surprisingly so. Different demographic and historical factors need to be considered to understand the genetic structure of extant populations [51].

### Effective family size and Québec evolutionary dynamics

Our study finds that the effective family size, EFS, remains a major player in the evolution of human populations. EFS represents the average number of children likely to increase offspring propagating generations down. The $SR_i$ declines in lockstep with a decreased EFS (Figs 2 and 3), revealing a very close correlation between the EFS and their $SR_i$ (Fig 3A and 3B). This correlation remains straightforward from the colony start until the end of the 19[th]-century: EFS determines the demographic outcome for many generations ahead.

The correlation between EFS and $SR_i$ is not fortuitous. It reflects the fate of the introduced lineages that can be readily followed in the $dc_i$ and $a_i$ plots (Fig 1E and 1F). These plots reveal the interplay between prosperous and extinct lineages, respectively. The immigrants $a_i$ soars dramatically at the turn of the 19[th] century, whereas their $dc_i$ drops (Fig 1E). The immigrants became less and less successful during the early 19th century, and their number Ni eventually exceeded their 1931–60 descendants ($Nw_i$). This starts when the $a_i$ and $dc_i$ plots cross when $Nw_i = Nc_i$. After 1850, immigrants' $dc_i$ increases and $a_i$ plateaus, but the balance is negative given $N_i > Nw_i$ (Fig 1E). Among QB, there is less variance in the corresponding plots (Fig 1F). QB $dc_i$ falls at the turn of the 18[th] century (matching that of the immigrants), levels off after this time and eventually rises during the 19[th] century (Fig 1F). QB $a_i$ never tops QB $dc_i$, yet the portion of their extinct lineages steadily rises, with $a_i$ noticeably increasing by the end of the 18[th]-century (Fig 1F, but see also Fig 3C and 3D).

Favored were immigrant parents who left plentiful married progeny. EFS, a fitness proxy, appears as the most important variable determining parental demographic success through future generations [11]. However, for this success to persist, the QB progeny must continue to be reproductively successful as overall SR is ultimately a function of successive EFS among the descendants. And indeed, a correlation of family size between generations was described in various populations [8,11,12,33], including species other than humans [11]. The reproductive success appears to be heritable, as shown in Québec and other preindustrial societies [6,13,43,52]. This might imply local selection [51] acting on new mutations and/or standing variation [53–55]. High EFS creates a surplus of new combinations among existing variants and increases new haplotypic arrangements. Non-recombining haplotypes of mtDNA and Y-chromosome could partake in these new combinations. High EFS continued over many generations suggesting that some of these combinations may present increased adaptive values and were favored [6,43]. By such traits, we understand biological (evolutionary), social, cultural, and economic benefits conducive to population expansion. In Québec, despite harsh environmental conditions, it seems to include many winning factors supporting survival.

### Factors affecting EFS estimates in Québec

We cannot ignore couples who failed to have children for various biological reasons, such as when one of the partners was sterile [56,57]. Sterility affects about 15% of contemporary couples [58] and was estimated to be about 10% in historical times. Therefore, the sterility of one of the partners must have contributed to the "zero fertility" peak in plots of the number of effective children that was published previously [6,33]. The overall childlessness reported in our plots corresponds to a cumulative effect of biological (see above) and other reasons that results in effective children being absent in the BALSAC records. If no married progeny is found in the BALSAC database, it can mean that the parents did have children, but these children died before reaching reproductive age; they did not marry or move outside of Québec [31,59]. If the childlessness were only due to biological causes, we would expect this to be relatively even among different Québec regions. However, we observed that zero peaks (no

children among married couples) markedly differed among the regional populations, and often, this difference exceeded 10–15% (S9 Fig).

Furthermore, the proportion of parents with no recorded children is more significant among immigrants than among QB (S10 Fig). This is consistent with the emigration of the first-generation immigrants or their children, who moved out of Québec [60] and disappeared from the BALSAC horizon (Fig 1E and 1F). In early colonial times, it also happened when some immigrant settlers moved outside Québec borders southwards to the United States or western Canada [31,59]. This phenomenon became exceptionally substantial in the 19th-century, engulfing many QB [59–61]. Emigration was motivated by unsatisfactory local conditions and tempting opportunities to prosper elsewhere. Local economic and social conditions started to worsen due to the rise in population density, military conflicts, famines, and epidemics [31,39,62,63]. This emigration affected the newcomers considerably more than the previous settlers. Therefore, when immigrants and their children left Québec, then "lost or extinct" does not necessarily imply that lineages absent from BALSAC records did not prosper elsewhere. In our calculations, we count only those that remained within the Québec borders.

Québec population was not the first colony benefitting from over-Atlantic European expansion. Europeans re-colonized different areas of the globe in historical times, often founding new nation-states, currently populated with their descendants, with various demographic and genetic outcomes. In Meso- and South America, a skewed sex ratio among European invaders, led to an asymmetric male-biased admixture with Native populations. European Y-chromosomes and Native mtDNA prevail in descendants [64–67]. Also, in Quebec, during the 17th-century colonization of the St. Lawrence Valley by the French, male settlers were more numerous than female immigrants [39]. Genetic studies indicate that at that time, admixture with Natives was limited [68], agreeing with Catholic Church records. Only a few Native females contributed to the French-Canadian genealogy [7,40,69]. Note, however, that many mixed marriages went presumably unrecorded, explaining thus the shortage of EFS-sons at the early colony years (S6 Fig). BALSAC database covers most Québec genealogies, but not all Québec diversity. Missing information includes some unnoted Natives and many Protestant Church records.

## Québec population history and founder events

In genetic studies of the Québec population, it was usually presumed that French-Canadians descended from about 8,500 immigrants [17] who settled in Nouvelle France before the British conquest in 1760 (e.g., [10,33,35,38,39]. We found that this premise is only partially correct. We show (Figs 1 and 2; Tables 1 and 2) that immigrants after the 1750s added many new matrilineal and patrilinear lines that were still present in the 1931–60 genetic pool. They exceeded the contribution of the first settlers in terms of the number of new lines but contributed much less to the mass of the genetic make-up of the post-1930 population (Table 2 and S5 Fig). And indeed, there is a striking difference between the 17th and the early 19th-century immigration. The population of the first immigrants was thrifty and thriving, populating a sparsely inhabited territory. In contrast, the subsequent immigrants were numerous, but their immigration success declined: the territory was already packed, and many immigrants failed to settle. It could be that during the 17th century, there was more social and economic cooperation, despite men's rivalry to find a spouse [17,39]. Although there were local conflicts with the Natives and the British, the environment was challenging but healthy since epidemics arrived only by the late 17th-century [63]. Therefore, being first and successful would have given these individuals the edge over subsequent generations in a quickly growing population [6,70,71]. The earliest immigrants benefited from demographic advantage reproducing when the population was small and given their EFS, they boosted their genetic representation in future

generations relative to latecomers [43,70,72,73]. In the early 19[th] century, Québec was already occupied by the well-established and prospering QB individuals. For newcomers, a suitable economic niche was more challenging to find. It persuaded many immigrants to temporarily settle in Québec and eventually move elsewhere [59]. Their EFS was dropping. Also, QB families were affected, albeit less dramatically. The fraction of reproductively prosperous families decreased (Fig 2). Interestingly, however, this occurred not everywhere, since the late 19[th]-century population of Saguenay–Lac-Saint-Jean was demographically as successful as that of the 17[th] century Québec settlers, where again the environment was harsh and challenging [6,74].

Québec colonization can be described as a series of successive founder events. It was accompanied by the founding of genetic mutations causing regionally distributed rare and endemic hereditary diseases ("medical founder effects"), well-documented in the Saguenay-Lac-St.-Jean and Charlevoix regions [73–77]. However, besides this medical interest, Québec also provides a founder population model in terms of the original concept developed by Ernst Mayr [78]. He stated that a relatively small group of migrants establishes a new population. Due to a sampling process, the overall diversity was reduced, and many variants were moved to different classes of the allelic frequency spectrum (S7 Fig). Likewise, while many rare variants were left behind, some, including defective, disease-causing alleles, could locally increase in frequency, thereby explaining "medical founder effects" [70,75,79,80]. Familial bonds led to non-random migration, when an individual would leave France to immigrate to Québec where his sibling had already settled [29], or when deported Acadians returned to New France to reunite with their families [61]. Such immigration bias could have favored the establishment of particular lineages. Sex-bias and founder effects may be typical to expanding human populations settling in an empty territory or conquering already occupied areas [81,82]. Because population dispersals occurred multiple times in the history of the human species [83], the question is to what extent Quebec demography, at a scale of fewer than 20 generations, may serve as an example of how different human populations evolved? A recent study describes the peopling of the Pacific Islands [84] that occurred over a similar time frame as the European colonization of the Americas. In Saint-Lawrence Valley, the population expansion went through a serial founder effect following a step-wise occupation of new territories. Likewise, the first arrivals from the Asiatic mainland to isolated Polynesian islands have experienced rapid initial growth, continued through a succession of founding bottlenecks, with resulting populations dominated by the genetic contribution of their founders [84]. We may ask weather Québec history provides a good model to analyze genetic diversity of expanding human populations undergoing founder effects, or presents only its particular example. We note, however, that founder effects are not reserved to the expanding humans and are also observed in other species [85].

Humans tend to keep contact with their founder and neighboring populations, even when separated by miles of ocean water. New immigrants constantly enrich the Quebec population. Tracing their origins should add to the genealogically inferred population diversity, especially when accompanied by population-wide genomic screening programs [86,87]. Such efforts help to understand better the history of the Québec population and its changing structure, assisting important epidemiological studies that impact health care programs.

## Supporting information

**S1 Fig. An example of matrilineal and patrilineal founders, immigrants, and Quebec-born individuals.** We define immigrants as the first generation of settlers to marry in Quebec (symbols circled in yellow; maternal immigrant in example I). Although strictly speaking, some of these immigrants were born in Quebec to immigrant parents. Immigrants can be lineage founders (symbols circled in a yellow frame filled with violet, example II). Genealogically recorded

non-immigrant lineage founders are marked in violet. If their immigrant progenitors are the sole introducers of these lineages, as shown in example I. Their genealogical impact is equivalent to that of solitary immigrants, as shown in example II. Lineage founders who are ancestors of more than one immigrant represent the same lineage introduced more than once. This may be the case of siblings who immigrated together (example III) or with distantly related cousins (example IV). Whenever the genealogical information permits, we avoid counting more than once the same patrilineal or matrilineal lineage introduced by immigrant siblings or immigrant cousins. Some immigrant lineages do not contribute to present-day Quebec diversity due to extinction by emigration (example I); by lack of daughters and/or sons altogether (examples III and IV, red-filled symbols). For the detailed description of data see: [19].
(DOCX)

**S2 Fig. Estimates of the Quebec population growth rate.** Growth rate ($k$) is estimated from the slope of the plot of historical population size *versus* time. In (**A**), we used population size data reported by Larin (red circles) ([31] and references therein) and Statistics Canada census data since 1851 (green circles) (http://www.stat.gouv.qc.ca/default_an.html). In (**B**), to estimate $k$, we used the yearly number of BALSAC recorded marriages (blue line–shown in pale blue in (**A**)) as a proxy for the population size. Linear regression curves were calculated by fitting the log of population sizes to the years by periods: 1621–1670 in dark green, 1671–1850 in brown and 1851–1960 in cyan. The associated slope coefficients ($k$) are displayed using the same color code.
(DOCX)

**S3 Fig. Actual numbers of 1931–60 descendants from immigration waves since 1621 compared to their simulated numbers assuming a constant survival ratio of 0.45 and 0.55, respectively, and constant annual growth rate ($k$) of 0.021 and 0.022 for male and female lineages, respectively.** In (**A**), paternal lineages with the observed (blue) and simulated (green) numbers of male descendants. (**B**) Maternal lines with the observed (red) and simulated (green) numbers of female descendants.
(DOCX)

**S4 Fig. Correlation between the effective family size and the survival ratio, as in Fig 3 in the main text, here based on 1-year intervals.** Overlapping plots of total EFSs and EFSd and their survival ratios for the immigrants (**A** and **B**) and the Quebec-born lineages (**C** and **D**). Their correlation plots are shown in the insets.
(DOCX)

**S5 Fig. Man to women sex ratio among the immigrants.** The number of male immigrants divided by the number of female immigrants considering all immigrants (**left panel**) and only a subset of the contributing ones (**right panel**).
(DOCX)

**S6 Fig. Comparing EFS-sons (EFSs) and EFS-daughters (EFSd) of all marrieds.** Please, note much lower EFS-sons (EFSs) than that of EFS-daughters (EFSd) before 1700, at the second half of the 17[th] century. This is consistent with the scenario of many Québec-born men, from immigrant and non-immigrant parents at that time, to leave Québec exploring other territories referred to as Nouvelle France.
(DOCX)

**S7 Fig. Frequency spectra of maternal and paternal lineages from before 1931 in the 1931–60 population.** The lineages introduced before 1801 are shown in the left panels, and those introduced between 1801 and 1930 are in the right plots. Note that on the y-axis, we present

the whole number of individuals within the frequency class to make the plot more informative and transparent as in (41). A typical plot of frequency classes registers only the number of classes on the y-axis. As a result, frequency classes represented only a few times or once, practically disappear from graphic representation, especially when frequency classes on the left are particularly numerous.
(DOCX)

**S8 Fig. EFS-sons and EFS-daughters of the contributing lineages introduced at different periods.** EFS is averaged over 10-year intervals, and progeny of lineages at different periods are marked by different colors as indicated. Remember that contributing refers to lineages still present within the 1931–60 Québec population.
(DOCX)

**S9 Fig. Distribution of the number of married children (total EFS) in families from different regions of Quebec.** Because only children married in Québec (registered in BALSAC) are counted, it emphasizes the potential effect of emigration out of Québec on the regional EFS frequency distribution. Note that "zero frequency EFS" exceeds the 10–15% threshold well, ascribed to couple sterility in many regions. Likewise, convex histograms of subsequent frequency classes, such as in Charlevoix, become concave when preceded by high zero EFS frequency, as in Outaouais or Isle de Montréal. Emigration affects the EFS of all families. Besides emigration (strictly, leaving Québec and married or not elsewhere), other factors: Economic, social, and historical (e.g., wars, epidemics) could have reduced BALSAC recorded EFS.
(DOCX)

**S10 Fig. Proportion of different categories of parents with no married children, i.e., of zero EFS.** Percentages of immigrant men (dark grey line), immigrant women (yellow line), QB men (cyan line), QB women (green line), all men (blue line), and all women (red line) without married children as a function of their marriage year by 5-year intervals. The light grey line indicates an estimate of the infertility rate (15%; [58]).
(DOCX)

## Acknowledgments

We are indebted to Claudia Moreau and Jean-Francois Lefebvre, who conducted initial analyses of the BALSAC data, and especially to Eef Harmsen, as well as to Jerzy Kolasa and Brad Loewen, for their comments on the manuscript. We are also thankful to members of the BALSAC team for their help in exploring the database.

## Author Contributions

**Conceptualization:** Damian Labuda.

**Data curation:** Tommy Harding, Hélène Vézina.

**Formal analysis:** Damian Labuda, Tommy Harding.

**Funding acquisition:** Damian Labuda.

**Investigation:** Damian Labuda.

**Methodology:** Damian Labuda.

**Project administration:** Damian Labuda.

**Software:** Tommy Harding.

**Supervision:** Damian Labuda.

**Validation:** Damian Labuda, Emmanuel Milot, Hélène Vézina.

**Visualization:** Damian Labuda.

**Writing – original draft:** Damian Labuda.

**Writing – review & editing:** Damian Labuda, Tommy Harding, Emmanuel Milot, Hélène Vézina.

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
