## [Decision Letter · Decision Letter 0]

4 Oct 2021

PONE-D-21-25384The effective family size of immigrant founders predicts their long-term demographic outcome: from Québec settlers to their 20th-century descendants.PLOS ONE

Dear Dr. Labuda,

Thank you for submitting your manuscript to PLOS ONE. After careful consideration, we feel that it has merit but does not fully meet PLOS ONE’s publication criteria as it currently stands. Therefore, we invite you to submit a revised version of the manuscript that addresses/discusses all the points raised during the review process. In particular, both Reviewers asked for a reassement of the interpretation about the uniparental markers. Please also ensure that your data will be fully available.

We look forward to receiving your revised manuscript.

Kind regards,

Alessandro Achilli, Ph.D.

Academic Editor

PLOS ONE

Journal Requirements:

Reviewers' comments:

Reviewer's Responses to Questions

**Comments to the Author**

1. Is the manuscript technically sound, and do the data support the conclusions?

Reviewer #1: Partly

Reviewer #2: Partly

2. Has the statistical analysis been performed appropriately and rigorously? 

Reviewer #1: I Don't Know

Reviewer #2: Yes

3. Have the authors made all data underlying the findings in their manuscript fully available?

Reviewer #1: Yes

Reviewer #2: No

4. Is the manuscript presented in an intelligible fashion and written in standard English?

Reviewer #1: Yes

Reviewer #2: Yes

5. Review Comments to the Author

Reviewer #1: To the authors:

The manuscript "The effective family size of immigrant founders predicts..." by Labuda et al. has some great potential considering the uniqueness of the Quebec population history and the availability of the BALSAC registry. For those that are greatly interested in both the genealogical and genetic aspects of population and individual histories, I find manuscripts such as this one of great value.

However, I am somewhat confused about the "genetic side" of the paper. In several instances there are references to the uniparental markers mitochondrial DNA and Y chromosome as if the paper would eventually discuss or correlate them with the available genealogical records. The authors are correct in explaining that genealogical studies are important in order to consider and predict genetic outcomes. However, they fail to tell us how this specific project would be relevant to Ycs and mtDNA studies. As a population is studied from the perspective of its founders and moving to the present time, numberless genealogical lines along not only the Ycs and mtDNA inheritance patterns, but especially concerning autosomal markers would arise and survive. Why would uniparental lines be of specific interest over autosomal ones?

Naturally, to expect that DNA data would be correlated to the Quebec population and the BALSAC database, both from the past (ancient DNA from burials) or modern DNA (from living individuals identified through descendancy research) would be an enormous effort, which would require not only a tremendous amount of resources, but would also need to address all sort of ethical research requirements. Therefore, I am wondering what the authors are suggesting by indicating that studies like the ones they have described in their manuscript have relevance with regard to large scale genetic studies, particularly with regard to uniparental markers.

Understanding the PLOS One is a multidisciplinary journal that does not focus on genetic studies, perhaps it would be better if the author would remove references to uniparental markers and focus on the relevance and findings of analyzing the BALSAC dataset.

This study is very similar to what Helgason et al. published in 2003 about the population of Iceland (reference 5 in the manuscript). The population isolate of Quebec and Iceland together with 300 years of documented genealogical data controlling for demographic events, including immigration brings these two studies very close together. Besides the difference in geographic location and in the dataset, there might be little to learn from a scientific viewpoint to make this a unique study. However, it might be worthwhile to some to see that a similar work was produced as some sort of second "testimonial" on the value of such datasets. My suggestion would be to look closely at how Helgason et al. combine the available genealogical data to other scientific aspects, including their correlation to uniparental markers. Additionally, it would be helpful (since Helgason was published nearly 20 years ago) to include something about what has changed in our understanding as a scientific community focusing on this types of studies over the past two decades.

There are a few additional minor points that have to do with the text that are enclosed as an attachment.

Reviewer #2: As a population geneticist it was a pleasure to read the paper written by Damian Labuda and colleagues about the effective family size of immigrants in Québec. This study is, however, more in the field of historical demography and evolutionary demography, rather than population genetics. Therefore, my comments are limited to the aspects which fit my expertise.

Major comment 1 = The authors have studied effective family sizes and long-term demographic outcomes among the Québec settlers solely (!) by focusing on direct paternal and maternal lineages which corresponds with the Y-chromosomal and mitochondrial DNA lineages. This ‘haploid’ view on the population history is of course highly limited if you consider all possible lineages in the whole genealogy. However, the abstract and discussion of the paper claims that this analysis gives information about the global (!) genetic structure of the founder population of Québec. For a long time, Y-chromosomal and mitochondrial DNA analyses were indeed very popular in human population genetics. Nowadays this has been changed drastically as we understand the strong limitations of haploid markers to study the population history, what makes whole-genome analyses the current reference for population genetics. Therefore, my main major comment on this paper is that the authors have to downsize at least their interpretation about the genetic structure of the founder population (especially in the abstract and discussion) as their analysis only provides information about the direct paternal and direct maternal lineages. They also have to persuade geneticists that there is a scientific relevance of studying only those two lineages instead of the whole genealogy to give satisfying answers on their general research questions related to the population genetics. From a genetic point of view they should include all genealogical lineages (for which they have even the data).

Major comment 2 = The authors focus on the paternal and maternal lineages on paper to discuss biological and genetic characteristics within the population across three centuries. However, the paternal lineage based on archival documents is biologically the most uncertain one due to the occurrence of nonpaternity. The authors have to discuss this issue broadly and guarantee that this nonpaternity issue has a limited influence on their interpretation for the Y-chromosomal variation in the Québec population, e.g. by refering to data from other human populations for which legal/documented versus biological/genetic paternity has been studied before.

Major comment 3 = The immigration history of the Québec population is very specific and almost unique in the human species. Therefore it is rather peculiar that the authors suggest that their analysis provides general insights for the human species (see discussion). I am not convinced that this can provide such insights. On the other hand their analysis can be highly useful in research on other species for which similar migration patterns occurs/occurred, like for example introduced species. It is clear that the authors have to look to the literature of introduced species which had often as well several immigration phases. As such they need to discuss if their observations according to evolutionary fitness within the Québec immigrant population has been observed within other species as well.

Major comment 4 = According to the PLOS policy the authors have to make all data underlying the findings in their manuscript fully available. This is not the case here. As most of the data is about historical periods and privacy is not an issue for historical data, there has to be a clear reason why it is not possible to provide all data (at least all data older than a certain age).

6. PLOS authors have the option to publish the peer review history of their article (what does this mean?). If published, this will include your full peer review and any attached files.

Reviewer #1: No

Reviewer #2: No

---

## [Author Response · Author response to Decision Letter 0]

12 Jan 2022

Answers to reviewer questions will be preceded with Au: will refer to important critical remarks and refer to the introduced changes in the article:

Comments to the Author 

3. Have the authors made all data underlying the findings in their manuscript fully available?

Reviewer #1: Yes

Reviewer #2: No

Au: Our mistake was not to mention that BALSAC data are fully available; however, their usage needs approval due to ethical concerns. The original data used in this study were obtained from BALSAC and deposited in Dataverse, as stated at the end of the text. 

It is possible that there will be some delay in the data appearance in Dataverse due to delays in formatting (in R)

The way the data were collected is described in (Vézina and Bournival 2020), where one finds a full description of the BALSAC database, its structure and how the data were gathered and validated.

Reviewer #1: To the authors:

The manuscript "The effective family size of immigrant founders predicts..." by Labuda et al. has some great potential considering the uniqueness of the Quebec population history and the availability of the BALSAC registry. For those that are greatly interested in both the genealogical and genetic aspects of population and individual histories, I find manuscripts such as this one of great value.

However, I am somewhat confused about the "genetic side" of the paper. In several instances there are references to the uniparental markers mitochondrial DNA and Y chromosome as if the paper would eventually discuss or correlate them with the available genealogical records. The authors are correct in explaining that genealogical studies are important in order to consider and predict genetic outcomes. 

However, they fail to tell us how this specific project would be relevant to Ycs and mtDNA studies. As a population is studied from the perspective of its founders and moving to the present time, numberless genealogical lines along not only the Ycs and mtDNA inheritance patterns, but especially concerning autosomal markers would arise and survive. Why would uniparental lines be of specific interest over autosomal ones?

Answer: 

Au: We may put too much emphasis on the "genetic" aspect of our study. And we hope to have corrected it in the Abstract and the Introduction (and Discussion), focusing on BALSAC resources and Québec population history. Here, uniparentally inherited lines provide the simplest picture of the population's genealogies, in contrast to autosomal lines that are particular to a given locus (e.g., see ref 25 – Nelson et al). In contrast to autosomal lineages, uniparentally inherited lineages and maternal and paternal genealogical lines follow the same pathways and are equivocal when representing families' genealogical and genetic history. The genealogical connection of uniparentally inherited lines to maternal mitochondrial DNA and paternal Y-chromosome lineages is obvious and straightforward. We do not have to use probabilistic approaches such as "gene dropping" (see ref 11 – Chen et al.). In other words, focusing on autosomal loci would be a subject of another study.

Au: While investigations of maternal and paternal lineages and interpretation of the results in the context of the mt and Y-chromosome inheritance are justified on their own merits, the question of how uniparental contribution relates to the autosomal outcomes of the contributing immigrants is absolutely pertinent. Given the reviewer comments, to bridge the contribution of the maternal-only and paternal-only lines to the actual (1931-60) population, we calculated the genetic contribution of the immigrant founders through autosomal pathways. We added a figure (Fig 5) showing this result. The relative contribution of uniparentally inherited female and male lines mirror that of the average contribution, via autosomal transmission, of the corresponding female and male founders. 

Au: As for the genealogical lines, we emphasize that going back in time from the present-day, as in the coalescent, we only learn about surviving lineages (ascending lines). Using in addition to surviving also extinct genealogical branches, we provide a fuller picture of the Quebec population history, presenting the fate of the consecutively arriving immigrants and the established Québec-born individuals. 

REV: Naturally, to expect that DNA data would be correlated to the Quebec population and the BALSAC database, both from the past (ancient DNA from burials) or modern DNA (from living individuals identified through descendancy research) would be an enormous effort, which would require not only a tremendous amount of resources but would also need to address all sort of ethical research requirements. Therefore, I am wondering what the authors are suggesting by indicating that studies like the ones they have described in their manuscript have relevance with regard to large scale genetic studies, particularly with regard to uniparental markers.

Answer: 

Au: Here again, thanks to reviewer comments, we added information on how uniparentally inherited markers were used to infer populations' history, at a very local up to worldwide scale, to infer sex-biased migrations, sex-biased reproduction, matrilocality and patrilocality and to reveal mechanisms shaping population structure and related genetic diversity. We also refer to a large genomic study involving genealogical records of the participants of CartaGene project in Québec (refs 86, 87) https://www.cartagene.qc.ca/en/about

REV: Understanding the PLOS One is a multidisciplinary journal that does not focus on genetic studies, perhaps it would be better if the author would remove references to uniparental markers and focus on the relevance and findings of analyzing the BALSAC dataset.

Answer: 

Au: One does not preclude the other. Because PlosOne is a multidisciplinary journal, its platform allows us to mix genetics and demography, showing how both of these are related in practice should be of general interest. We feel that disregarding genetic systems associated with uniparentally inherited genealogical lines, in this situation, would eliminate this important message we focus on in our paper. But again, we agree with the reviewer that our message required re-equilibration (as described above).

REV: This study is very similar to what Helgason et al. published in 2003 about the population of Iceland (reference 5 in the manuscript). The population isolate of Quebec and Iceland together with 300 years of documented genealogical data controlling for demographic events, including immigration brings these two studies very close together. Besides the difference in geographic location and in the dataset, there might be little to learn from a scientific viewpoint to make this a unique study. However, it might be worthwhile to some to see that a similar work was produced as some sort of second "testimonial" on the value of such datasets. My suggestion would be to look closely at how Helgason et al. combine the available genealogical data to other scientific aspects, including their correlation to uniparental markers. Additionally, it would be helpful (since Helgason was published nearly 20 years ago) to include something about what has changed in our understanding as a scientific community focusing on these types of studies over the past two decades.

Answer: 

Au: While, to some extent, Helgason's study inspired our investigations, exploiting the direct connection between the genetics of uniparentally inherited markers and the corresponding genealogical lines, our study is different in a qualitative and in a quantitative manner. Helgason's study relies on the ascending genealogies only, probing founders from two historical periods –the first half of the 18th and second half of the 19th century. In the text, we emphasized this difference. As for other studies linking genealogy and genetics, we also refer to papers from our laboratories and different research groups.

Au: Our study involves genealogies of a population of more than two million (in 1931-60) and covers data on about 3.3 million married Québec individuals. It presents a history of a different population and how consecutive immigration waves shaped it. The distribution of genetic contribution of uniparental lines introduced at different immigration waves mirrors that of their autosomal genomes contributions (no variance included). The reviewer's comments helped us clarify this information in the revised manuscript, keeping the message short and concise.

REV: There are a few additional minor points that have to do with the text that are enclosed as an attachment. – 

answered below:

Reviewer #2: As a population geneticist it was a pleasure to read the paper written by Damian Labuda and colleagues about the effective family size of immigrants in Québec. This study is, however, more in the field of historical demography and evolutionary demography, rather than population genetics. Therefore, my comments are limited to the aspects which fit my expertise.

Major comment 1 =

The authors have studied effective family sizes and long-term demographic outcomes among the Québec settlers solely (!) by focusing on direct paternal and maternal lineages which corresponds with the Y-chromosomal and mitochondrial DNA lineages. This 'haploid' view on the population history is of course highly limited if you consider all possible lineages in the whole genealogy. 

Answer: 

Au: While studying uniparentally inherited lineages is very important on its own, correlating their inheritance with that of the accompanying autosomes was never addressed. To answer the reviewer's concerns, we carried additional "experiments", estimating the autosomal genetic contribution of the immigrant founders, females and males" on our genealogies. The results are presented in new Fig 5 added to the original ms, as already described in the answers to the Reviewer 1.

REV: However, the abstract and discussion of the paper claims that this analysis gives information about the global (!) genetic structure of the founder population of Québec. For a long time, Y-chromosomal and mitochondrial DNA analyses were indeed very popular in human population genetics. Nowadays this has been changed drastically as we understand the strong limitations of haploid markers to study the population history, what makes whole genome analyses the current reference for population genetics.

 Therefore, my main major comment on this paper is that the authors have to downsize at least their interpretation about the genetic structure of the founder population (especially in the abstract and discussion) as their analysis only provides information about the direct paternal and direct maternal lineages. 

Answer: 

Au: We agree with the reviewer's critical comments, consistent with Reviewer 1. Our manuscript was modified accordingly as presented in our earlier answers, not to be repeated again here. 

REV: They also have to persuade geneticists that there is a scientific relevance of studying only those two lineages instead of the whole genealogy to give satisfying answers on their general research questions related to the population genetics. From a genetic point of view, they should include all genealogical lineages (for which they have even the data).

Answer: 

Au: There is no conflict among population geneticists, we believe, between those studying mtDNA and Y-chromosome lineages and those using whole-genome diversity data. These data are complementary and are usually used to answer different questions. Here, we also show that along uniparentally inherited lines contribution, we may follow the accompanying autosomal lineages (assuming neutrality and no selection) as reported in Fig 5. Binomial distribution of autosomal alleles among the progeny of both sexes and over many generations creates a genealogical network through which we can study particular loci. To focus on population history, we would have to average different possible pathways from founders down the generations, leading us to a result presented in Fig 5. We believe that our approach is simpler and straightforward. Importantly, this reviewer comments helped us discuss these problems in our revised version.

REV: Major comment 2 = 

The authors focus on the paternal and maternal lineages on paper to discuss biological and genetic characteristics within the population across three centuries. However, the paternal lineage based on archival documents is biologically the most uncertain one due to the occurrence of nonpaternity. The authors have to discuss this issue broadly and guarantee that this nonpaternity issue has a limited influence on their interpretation for the Y-chromosomal variation in the Québec population, e.g. by refering to data from other human populations for which legal/documented versus biological/genetic paternity has been studied before.

Answer :

Au: This is a legitimate concern, and we forget to mention this issue (discussed in our previous study by Harding et al. ref 26). Non-paternity is very rare even in the present-day French-Canadians, based on our anecdotal data from familial diagnostic studies we carried at HSJ in the 1980-90-is where none such case was detected. Serious scientific data are discussed in Harding et al, we cite when we discuss this issue: overall genealogical errors due to different factors, including nonpaternity, is estimated at below 1% (~0.75). These are very unlikely to affect our global results. 

REV: Major comment 3 =

 The immigration history of the Québec population is very specific and almost unique in the human species. Therefore, it is rather peculiar that the authors suggest that their analysis provides general insights for the human species (see discussion). I am not convinced that this can provide such insights. On the other hand, their analysis can be highly useful in research on other species for which similar migration patterns occurs/occurred, like for example introduced species. It is clear that the authors have to look to the literature of introduced species which had often as well several immigration phases. As such they need to discuss if their observations according to evolutionary fitness within the Québec immigrant population has been observed within other species as well. 

Answer:

Au: We agree with the reviewer but addressing all these issues requires additional papers on their own. However, we addressed some of these issues in our discussion. Concerning Quebec as a model of expanding population, we cited a recent study by Ioannidis et al on serial founder effect in peopling of Pacific islands (ref 84). We provided an example of other species founder effects by Clegg et al (ref 85). We also refer to a genealogical study concerning a population of a species of jaybirds in Florida (Chen et al ref 11). We believe that these examples show that Québec, with its particular aspects, is not unique in human populations' history. We discussed other studies pointing to the role of founder effects followed by expansions but not explicitly addressing this issue.

REV: Major comment 4 = 

According to the PLOS policy the authors have to make all data underlying the findings in their manuscript fully available. As most of the data is about historical periods and privacy is not an issue for historical data, there has to be a clear reason why it is not possible to provide all data (at least all data older than a certain age).

Answer:

Au: Our negligence - see our answer to this issue above (before particular answers to reviewers comments)

Plos One Review – minor comments Reviewer 1

Line 23 - Not sure what it is meant with "we followed their transmission in real-time" within this context. Please explain or rephrase. 

Au: We are very grateful to the reviewer for generously suggesting these minor corrections to improve our text and message. 

We corrected all, but due to subsequent modifications of the original text, citing these corrections becomes irrelevant because the text could have been changed. All were carefully introduced in our original text b

---

## [Decision Letter · Decision Letter 1]

15 Mar 2022

The effective family size of immigrant founders predicts their long-term demographic outcome: from Québec settlers to their 20th-century descendants.

PONE-D-21-25384R1

Dear Dr. Labuda,

We’re pleased to inform you that your manuscript has been judged scientifically suitable for publication and will be formally accepted for publication once it meets all outstanding technical requirements.

Kind regards,

Alessandro Achilli, Ph.D.

Academic Editor

PLOS ONE

Additional Editor Comments (optional):

Reviewers' comments:

Reviewer's Responses to Questions

**Comments to the Author**

1. If the authors have adequately addressed your comments raised in a previous round of review and you feel that this manuscript is now acceptable for publication, you may indicate that here to bypass the “Comments to the Author” section, enter your conflict of interest statement in the “Confidential to Editor” section, and submit your "Accept" recommendation.

Reviewer #1: All comments have been addressed

2. Is the manuscript technically sound, and do the data support the conclusions?

Reviewer #1: Yes

3. Has the statistical analysis been performed appropriately and rigorously? 

Reviewer #1: I Don't Know

4. Have the authors made all data underlying the findings in their manuscript fully available?

Reviewer #1: Yes

5. Is the manuscript presented in an intelligible fashion and written in standard English?

Reviewer #1: Yes

6. Review Comments to the Author

Reviewer #1: To the authors. I have no further comments and I am satisfied on how my review has been addressed. Table 2 did not come through properly formatted on the pdf file and it looks like there might be more data that I was not able to view. Also there is a comment on the right margin that might have been left there from an internal review while addressing the reviewers comments.

7. PLOS authors have the option to publish the peer review history of their article (what does this mean?). If published, this will include your full peer review and any attached files.

Reviewer #1: No

---

## [Editor Report · Acceptance letter]

7 Apr 2022

PONE-D-21-25384R1 

The effective family size of immigrant founders predicts their long-term demographic outcome:  from Québec settlers to their 20th-century descendants. 

Dear Dr. Labuda:

I'm pleased to inform you that your manuscript has been deemed suitable for publication in PLOS ONE. Congratulations! Your manuscript is now with our production department. 

Kind regards, 

on behalf of

Prof. Alessandro Achilli 

Academic Editor

PLOS ONE